# New Chlorinated Metabolites and Antiproliferative Polyketone from the Mangrove Sediments-Derived Fungus *Mollisia* sp. SCSIO41409

**DOI:** 10.3390/md21010032

**Published:** 2022-12-30

**Authors:** Jian Cai, Xueni Wang, Xia Gan, Qian Zhou, Xiaowei Luo, Bin Yang, Yonghong Liu, Disna Ratnasekera, Xuefeng Zhou

**Affiliations:** 1CAS Key Laboratory of Tropical Marine Bio-resources and Ecology, Guangdong Key Laboratory of Marine Materia Medica, South China Sea Institute of Oceanology, Chinese Academy of Sciences, Guangzhou 510301, China; 2University of Chinese Academy of Sciences, Beijing 100049, China; 3Guangxi Zhuang Yao Medicine Center of Engineering and Technology, Guangxi University of Chinese Medicine, Nanning 530200, China; 4Institute of Marine Drugs, Guangxi University of Chinese Medicine, Nanning 530200, China; 5Department of Agricultural Biology, Faculty of Agriculture, University of Ruhuna, Matara 81000, Sri Lanka

**Keywords:** mangrove sediments-fungus, *Mollisia* sp., chlorinated metabolites, prostate cancer, antimicrobial

## Abstract

Two new chlorinated metabolites, 8-chlorine-5-hydroxy-2,3-dimethyl-7-methoxychromone (**1**) and 3,4-dichloro-1*H*-pyrrole-2,5-dione (**3**), and eight known compounds (**2** and **4**–**9**) were isolated from the mangrove sediments-derived fungus *Mollisia* sp. SCSIO41409. Their structures were elucidated by physicochemical properties and extensive spectroscopic analysis. The absolute configuration of stemphone C (**4**) was established for the first time by the X-ray crystallographic analysis. Compounds **3** and **4** showed different intensity of antimicrobial activities against several pathogenic fungi and bacteria, and antiproliferative activities against two human prostate cancer cell lines (IC_50_ values 2.77 to 9.60 μM). Further, stemphone C (**4**) showed a reducing PC-3 cell colony formation, inducing apoptosis and blocking the cell cycle at S-phase in a dose-dependent manner; thus, it could be considered as a potential antiproliferative agent and a promising anti-prostate cancer lead compound.

## 1. Introduction

Mangrove forests are a complex ecosystem, and the microbes in mangrove sediments play an important role in the biogeochemical cycles of mangrove ecosystems [1]. The high diversity of mangrove environments has contributed to high microbial diversity, which is an important source of bioactive natural products [2,3,4]. Research on secondary metabolites from mangrove sediments-derived microbes has yielded many natural products with novel structures and significant pharmacological activities [5,6,7,8]. In particular, halogenated compounds obtained from mangrove-derived microbes, especially chlorine-containing metabolites, have received a great deal of attention [9,10,11].

*Mollisia* is a taxonomically neglected discomycete genus (Helotiales, Leotiomycetes) in decaying plant tissues or root soil. The natural products from the genus *Mollisia* were not well explored and were limited to few publications. Mollisinols A and B, two new metabolites were isolated from the endophytic fungus *Mollisia* sp., derived from the root bark of *Ardisia cornudentata* Mez [3]. Ophiobolin C, isolated from *Mollisia* sp. (GB5328) from the dead bark of *Tsuga canadensis*, was an inhibitor of binding with human CCR5 receptor and exhibited an IC_50_ value of 40 μM [12]. KS-504 compounds, three novel inhibitors of Ca^2+^ and calmodulin-dependent cyclic nucleotide phosphodiesterase were isolated from *Mollisia ventosa* KAC-1 148 [13]. Mollisianitrile, a new antibiotic was isolated from *Mollisa* sp. A59–96 [14]. Benesudon, isolated from *Mollisia bensuada*, possessed antimicrobial, cytotoxic, and phytotoxic activities [15]. Mollisin, a dichloronaphthoquinone derivative produced by the fungus *Mollisia caesia* [16]. In our study, we explored potential bioactive secondary metabolites from a mangrove sediment-derived fungal strain, *Mollisia* sp. SCSIO41409, isolated from a mangrove sediment sample in Hainan Island, China. During our search for potentially diverse and bioactive secondary metabolites from mangrove fungal sources [10,17,18,19], two new chlorinated metabolites (**1** and **3**), and seven known polyketides (**2** and **4**–**9**) (Figure 1) were isolated and identified from a mangrove sediment-derived fungus *Mollisia* sp. SCSIO41409. These compounds were examined for antimicrobial and antiproliferative activities. Herein, we report the details of the isolation, structural elucidation, and biological evaluation of all isolated compounds.

## 2. Results

### 2.1. Structural Determination

Compound **1** was isolated as a white needle crystal and had the molecular formula C_12_H_11_ClO_4_ as determined by the HRESIMS spectrum, which showed a cluster of protonated ion peaks at *m*/*z* 255.0422/257.0394 [M + H]^+^ with the ratio of 3:1, indicative of a monochlorinated compound. The 1D NMR (Table 1) and HSQC spectrum of **1** showed signals of a carbonyl carbon (*δ*_C_ 181.1), seven non-protonated *sp*^2^ carbons (*δ*_C_ 163.9, 160.0, 159.9, 151.6, 114.6, 103.7 and 97.4), one aromatic methine (*δ*_H/C_ 6.62/95.9), one oxygenated methyl (*δ*_H/C_ 3.94/57.0), and two methyls (*δ*_H/C_ 2.42/18.3 and 1.91/8.8). The above-mentioned data combined with seven degrees of unsaturation suggested that **1** presented a chromone skeleton. The above NMR characteristics showed great similarity to those of the co-isolated **2**, which was reported as a chromone compound. The main difference was the presence of the chlorine atom instead of a hydrogen atom at C-8 in **1** and this deduction was supported by the above 2D NMR data. The HMBC correlations (Figure 2) from H_3_-9 to C-2 and C-3, from H_3_-10 to C-2, C-3, and C-4, revealed that the methyl groups CH_3_-9 and CH_3_-10 were located at C-2 and C-3, respectively. The HMBC correlations from 5-OH to C-4a, C-5, and C-6 indicated the location of the phenolic hydroxyl group (C-5). The chemical shift of C-8 (*δ*_C_ 97.4) revealed the substitution of chlorine instead of the oxygenated methyl, which was attached at C-8. The methoxy group was deduced to link with C-7 by the HMBC signal of H_3_-11/C-7 and the chemical shift of C-7 (*δ*_C_ 159.9). The X-ray crystal structure of **1** (CDCC 2221470), obtained by slow evaporation in CH_3_OH, further confirmed the above elucidation of the planar structure. Compound **1** was unambiguously characterized as shown in Figure 1 and defined as 8-chlorine-5-hydroxy-2,3-dimethyl-7-methoxychromone.

Compound **3** was isolated as a red needle crystal, and its molecular formula of C_4_HCl_2_NO_2_ was determined by HRESIMS data at *m*/*z* 163.9318 [M–H]^–^ (calcd for C_4_Cl_2_NO_2_, 163.9312). The analysis of its structure accurately from the NMR spectrum was difficult, because of its symmetrical structure and simplicity of carbons (*δ*_C_ 164.1, 132.8) and hydrogens (*δ*_H_ 11.71 (s, 1H)). However, we obtained X-ray crystal data (Cu K*α* radiation) of **3,** and the structure was accurately determined as 3,4-dichloro-1*H*-pyrrole-2,5-dione (Figure 3). Compound **3** has been previously identified as a synthetic product, which was synthesized to illustrate the insecticidal structure-activity relationship of the *N*-amino-maleimide derivatives [20]. Here we are reporting the first time this compound explored from nature, as a new metabolite or new natural product. 

Compound **4**, obtained as yellow needles, was found to have the molecular formula C_30_H_42_O_7_ on the basis of HRESIMS data at *m*/*z* 515.3003 [M+H]^+^ (calcd C_30_H_43_O_7_, 515.3003). The ^13^C NMR and DEPT data displayed 30 carbon signals including eight methyls, five methylenes, two *sp*^3^ methine, two *sp*^2^ methine, three oxygenated *sp*^3^, one *sp*^3^ quaternary, two oxygenated *sp*^3^ quaternary, four *sp*^2^ quaternary and three carbonyls. Detailed comparison of the above NMR data with the literature [21], **4** was identified as stemphone C. However, the absolute configuration of **4** had not been reported. We determined the absolute configuration of **4** as 4*S*, 5*S*, 13*R*, 14*R*, 17*R*, 18*R*, 21*R* (Figure 3) by X-ray crystallographic analysis using Cu K*α* radiation.

Meanwhile, the other seven known compounds were identified as 5-hydroxy-2,3-dimethyl-7-methoxychromone (**2**) [22], *cis*-cyclo (Tyr-Ile) (**5**) [23],4,8-dihydroxy-1-tetralone (**6**) [24], cyclo (Phe-Tyr) (**7**) [25], tenuissimasatin (**8**) [26], 4-methyl-5,6-dihydro-2H-pyran-2-one (**9**) [27], respectively, by comparison of their NMR data (Appendix A) with previous reports.

**Figure 2 marinedrugs-21-00032-f002:**
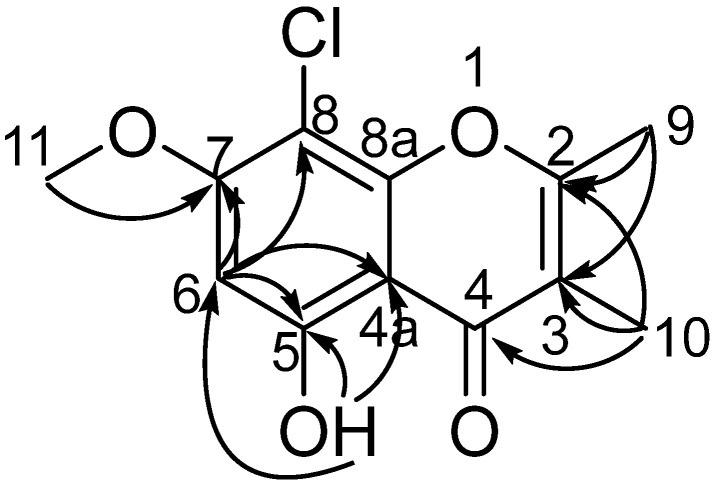
Key HMBC (arrows) correlations of **1**.

**Figure 3 marinedrugs-21-00032-f003:**
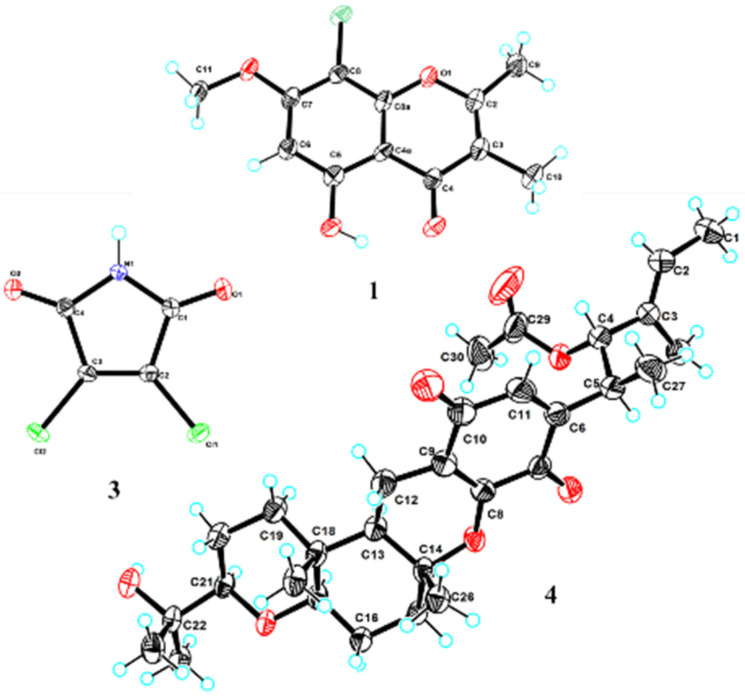
X-ray single-crystal structures of compounds **1**, **3**, and **4**.

**Table 1 marinedrugs-21-00032-t001:** The NMR data of compound **1** (500 and 125 MHz, *δ* in ppm, DMSO-*d*_6_).

Pos.	*δ*_C_ Type	*δ*_H_ (*J* in Hz)	HMBC
2	163.9, C		
3	114.6, C		
4	181.1, C		
4a	103.7, C		
5	160.0, C		
6	95.9, CH	6.62 (s)	4a, 5, 7, 8
7	159.9, C		
8	97.4, C		
8a	151.6, C		
9	18.3, CH3	2.42 (s)	2, 3
10	8.8, CH3	1.91 (s)	2, 3, 4
11	57.0, CH3	3.94 (s)	7
5-OH		13.09 (s)	4a, 5, 6

### 2.2. Antimicrobial and Antiproliferative Activities

All of the obtained compounds were evaluated for the activities of pathogenic fungi and bacteria commonly found in crop plants (Table 2). Compound **3** exhibited antifungal activities against *Botrytis cinerea*, *Verticillium dahlia* kieb., *Fusarium graminearum* schw., *Fusarium oxysporum* f.sp. niveum, *Rhizoctonia solani,* and *Septoria nodorum* Berk., with the MIC values of 25–50 μg/mL. Compound **4** exhibited antibiotic activity against bacteria *Erysipelothrix rhusiopathiae* WH13013 and *Streptococcus suis* SC19, with the MIC values of 1.56 and 6.25 μg/mL. In particular, the strength of **4** against *E. rhusiopathiae* was stronger than that of penicillin with the MIC value of 6.25 μg/mL.

Two human prostate cancer cell lines, PC-3 (androgen receptor negative) and 22Rv1 (androgen receptor positive), were used in the antiproliferative assay for all the obtained compounds. Compound **3** exhibited antiproliferative activity against 22Rv1 and PC-3 cells with IC_50_ values of 8.35 and 9.60 μM, respectively, while **4** showed activities against 22Rv1 and PC-3 cells with IC_50_ values of 5.81 and 2.77 μM, respectively. In order to evaluate whether **4** selectively inhibits prostate cancer cells, we screened other cancer cells for antiproliferative activity. The result showed that **4** was also active against other cells (HepG2, A549, Hela, WPMY-1, MC3T3-E1) with IC_50_ values of 3.63–11.68 μM. Thus, **4** had the most significant effect on PC-3 cells compared to other compounds with broader inhibitory activity.

To further evaluate the inhibitory effect of **4** on PC-3 cells, we performed a plate clone formation assay. The results showed that **4** significantly inhibited the formation of clonal colonies of PC-3 cells and its inhibitory effect was positively correlated with the dose (Figure 4). In addition, we examined the effect of **4** on PC-3 cell apoptosis by flow cytometry. It was revealed that **4** could significantly induce apoptosis in PC-3 cells. As shown in Figure 5, when PC-3 cells were treated with 10 μM of **4** for 48 h, 14.64% of the cells were induced to early apoptosis and 36.84% of the cells were induced to late apoptosis. This finding suggests that the induction of apoptosis in PC-3 cells is a mode of action for the production of antiproliferative activity by **4**.

To identify the inhibitory process of the proliferation of prostate cancer cells, we detected the cell cycle distribution of PC-3 cells. As shown in Figure 6, when treated with the M-phase blocker docetaxel, a large number of PC-3 cells were blocked in M-phase, with a dramatic increase in the ratio of G2/M, up to 71.23 percent. However, unlike in the case of docetaxel, the percentage of S-phase was significantly increased in cells treated with **4**. When the concentration of **4** reached 10 μM, the percentage of cells in the S-phase was as high as 59.14 percent. These results suggested that **4** blocked the cell cycle at S-phase, impairing cell proliferation. Consequently, it is revealed that **4** is a promising lead compound for pharmacotherapy of prostate cancer.

## 3. Discussion

Marine organisms and microorganisms living in extreme marine environments were considered to be important sources of halogenated compounds [28]. Chlorinated metabolites, as a key part of the halogenated compounds, have demonstrated a wide range of significant activities. Due to the high concentration of chloride ions in mangrove ecosystem, many chlorinated compounds have been excavated from mangrove-derived microorganisms [9,10,11]. In this study, two new chlorinated metabolites, **1** and **3**, were isolated from mangrove sediments-derived fungus. Although 3,4-dichloro-1*H*-pyrrole-2,5-dione (**3**) has been previously reported as a synthetic product, it was discovered as a new metabolite of natural origin in our study. So, mangrove sediments-derived microbes have proved to be a promising source of novel and unique chlorine-containing bioactive secondary metabolites.

An important result of this study is the discovery of stemphone C (**4**) with significant anti-prostate cancer activity in vitro. Stemphone C (**4**) had been reported from an *Aspergillus* strain as a potentiator of imipenem activity against methicillin-resistant *Staphylococcus aureus*, with other stemphone derivatives [21,29]. Stemphone was found to inhibit over contraction of portal vein, induced by high glucose levels [30], and its derivatives were also revealed to inhibit lipid droplet accumulation in macrophages [31]. In our study, stemphone C (**4**) exhibited obvious antimicrobial activity against the bacteria *Erysipelothrix rhusiopathiae* WH13013, with the MIC value of 1.56 μg/mL. In addition, obvious antiproliferative activities of stemphone C (**4**) were also revealed against two prostate cancer cell lines (PC-3 and 22Rv1) and three other cancer cell lines (HepG2, WPMY-1, and MC3T3-E1), with IC_50_ values less than 10 μM. Furthermore, it showed reducing PC-3 cells colony formation, inducing apoptosis, and blocking the cell cycle at S-phase in a dose-dependent manner. Our study showed that stemphone C (**4**) could be considered as a potential antiproliferative agent, especially as a promising anti-prostate cancer lead compound. Importantly, we could able to determine the absolute configuration of **4**, the first time in this study, which is necessary for further development of this active compound.

## 4. Materials and Methods

### 4.1. General Experimental Procedures

The UV and IR spectra were recorded on a Shimadzu UV-2600 PC spectrometer (Shimadzu, Beijing, China) and an IR Affinity-1 spectrometer (Shimadzu), respectively. Optical rotations were determined with an Anton Paar MPC 500 (Anton, Graz, Austria) polarimeter. High resolution electrospray ionization mass spectroscopy (HRESIMS) spectra were acquired on a Bruker maXis Q-TOF mass spectrometer (Bruker BioSpin International AG, Fällanden, Swizerland). The NMR spectra were recorded on a Bruker Avance spectrometer (Bruker) operating at 500 and 700 MHz for ^1^H NMR and 125 and 175 MHz for ^13^C NMR that used tetramethylsilane as an internal standard. Semipreparative high-performance liquid chromatography (HPLC) was performed on the Hitachi Primaide with a DAD detector, using an ODS column (YMC-pack ODS-A, 10 × 250 mm, 5 μm). Column chromatography was performed over silica gel (200–300 mesh) (Qingdao Marine Chemical Factory, Qingdao, China). Spots were detected on TLC (Qingdao Marine Chemical Factory) under 254 nm UV light. All solvents employed were of analytical grade (Tianjin Fuyu Chemical and Industry Factory, Tianjin, China).

### 4.2. Fungal Material

The fungal strain *Mollisia* sp. SCSIO41409 was isolated from a mangrove sediment sample, collected from the Hongsha River estuary near South China Sea, in Sanya city, Hainan Island. The fungus was identified according to the internally transcribed spacer (ITS) region sequence data of the rDNA (Appendix A), and the sequence was deposited in GenBank with the accession number OP872608. A voucher specimen was deposited in the CAS Key Laboratory of Tropical Marine Bioresources and Ecology, South China Sea Institute of Oceanology, Chinese Academy of Sciences, Guangzhou, China.

### 4.3. Fermentation and Extraction

The fungal strain was cultured in 200 mL seed medium (1.5% malt extract, 1.5% sea salt) in 500 mL Erlenmeyer flasks at 28 °C for 3 days on a rotary shaker (180 rpm). A large-scale fermentation was incubated statically at 26 °C for 60 days using a rice medium (150 g rice, 1.5% sea salt, 150 mL H_2_O) in the 1 L flask (×53). The whole fermented culture was extracted with EtOAc three times to afford a brown extract (197.2 g).

### 4.4. Isolation and Purification

The crude extract was chromatographed over a silica gel column eluted with petroleum ether /CH_2_Cl_2_ (0–100%, *v*/*v*) and CH_2_Cl_2_/CH_3_OH (0–100%, *v*/*v*) to obtain eleven fractions (Frs. 1–11) based on TLC properties. Fr. 2 was subjected to semipreparative HPLC (63%CH_3_CN/H_2_O, 2.5 mL/min) to afford **2** (3.7 mg, t_R_ = 18.6 min) and **1** (9.1 mg, t_R_ = 27.0 min). Fr. 4 was separated by semipreparative HPLC (40% MeCN/H_2_O, 2.5 mL/min) to afford **9** (7.1 mg, t_R_ = 6.5 min), **8** (4.1 mg, t_R_ = 8.1 min), **3** (27.8 mg, t_R_ = 11.1 min). Fr. 5 was separated by semipreparative HPLC (37% MeCN/H_2_O, 2.5 mL/min) to afford **6** (8.6 mg, t_R_ = 9.8 min). **4** (8.0 mg, t_R_ = 24.0 min) was obtained from Fr. 6 by semipreparative HPLC eluting with 76% CH_3_CN/H_2_O (2.5 mL/min). Fr. 10 was separated by semipreparative HPLC (23% MeCN/H_2_O, 2.5 mL/min) to afford **7** (22.0 mg, t_R_ = 19.0 min) and **5** (9.0 mg, t_R_ = 12.0 min). 

### 4.5. Spectroscopic Data of Compounds

8-chlorine-5-hydroxy-2,3-dimethyl-7-methoxychromone (**1**): white needles, m.p. 196–198 °C; UV (CH_3_OH) *λ*_max_ (log *ε*) 327 (3.54), 282 (3.49), 259 (4.10), 244 (4.15), 204 (4.11) nm; IR (film) *ν*_max_ 3734, 2926, 2855, 1717, 1653, 1558, 1437, 1207, 1182, 1103, 820 cm^−1^; ^1^H and ^13^C NMR data as shown in Table 1; HRESIMS *m*/*z* 255.0422 [M + H]^+^ (calcd for C_12_H_12_ClO_4_^+^, 255.0419).

3,4-dichloro-1*H*-pyrrole-2,5-dione (**3**): red needles, m.p. 173–175 °C; UV (CH_3_OH) *λ*_max_ (log *ε*) 235.60 (0.80), 229.20 (0.79) nm; IR (film) *ν*_max_ 3204, 1732, 1609, 1331, 1047, 1026, 851, 735, 673, 550 cm^−1^; ^1^H NMR (700 MHz, DMSO-*d*_6_) *δ* 11.71 (s, 1H); ^13^C NMR (175 MHz, DMSO) *δ* 164.06, 132.82. HRESIMS data at *m*/*z* 163.9318 [M–H]^–^ (calcd for C_4_Cl_2_NO_2_, 163.9312).

### 4.6. X-ray Crystallographic Analysis

The clear light colorless crystal of **1** was obtained in MeOH by slow evaporation. Crystallographic data for the structure has been deposited in the Cambridge Crystallographic Data Centre. Copies of the data can be obtained, free of charge, on application to CCDC, 12 Union Road, Cambridge CB21EZ, UK [fax: +44(0)-1223-336033 or e-mail: deposit@ccdc.cam.ac.uk].

Crystal data for **1**: 2C_12_H_11_ClO_4_, *M*r = 509.31, crystal size 0.3 × 0.03 × 0.02 mm^3^, triclinic, *a* = 10.3781 (6) Å, *b* = 10.7434 (9) Å, *c* = 11.7251 (8) Å, *α* = 94.077(6)°, *β* = 101.693 (5)°, *γ* = 118.572 (7)°, *V* = 1102.59 (14) Å^3^, *Z* = 4, *T* = 100.00 (10) K, space group *P*-1, *μ*(Cu K*α*) = 3.099 mm^−1^, *D_calc_* = 1.534 g/cm^3^, 3879 reflections measured (7.852° ≤ 2Θ ≤ 133.17°), 3879 unique (*R*_sigma_ = 0.0731). The final *R*_1_ values were 0.0733 (I > 2*σ*(I)). The final *wR*(F^2^) values were 0.2186 (I > 2*σ*(I)). The final *R*_1_ values were 0.1004 (all data). The final *wR*(F^2^) values were 0.2360 (all data). The goodness of fit on F^2^ was 1.059 (CCDC 2221470).

Crystal data for **3**: 2C_4_HCl_2_NO_2_, *M*r = 165.96, crystal size 0.09 × 0.06 × 0.06 mm^3^, monoclinic, *a* = 7.11690 (10) Å, *b* = 8.03280 (10) Å, *c* = 10.2446 (2) Å, *α* = 90°, *β* = 99.9790(10)°, *γ* = 90°, *V* = 576.809 (16) Å^3^, *Z* = 4, *T* = 100.00 (10) K, space group *P*2_1_/c, *μ*(Cu K*α*) = 9.446 mm^−1^, *D_calc_* = 1.911 g/cm^3^, 2500 reflections measured (12.63° ≤ 2Θ ≤ 148.6°), 1126 unique (*R*_sigma_ = 0.0272). The final *R*_1_ values were 0.0239 (I > 2*σ*(I)). The final *wR*(F^2^) values were 0.0666 (I > 2*σ*(I)). The final *R*_1_ values were 0.0246 (all data). The final *wR*(F^2^) values were 0.0671 (all data). The goodness of fit on F^2^ was 1.092 (CCDC 2221370).

Crystal data for **4**: 2C_30_H_42_O_7_·CH_3_OH, *M*r = 1061.31, crystal size 0.06 × 0.06 × 0.05 mm^3^, orthorhombic, *a* = 6.37690 (10) Å, *b* = 19.5291 (3) Å, *c* = 49.6907 (8) Å, *α* = 94.077(6)°, *β* = 101.693(5)°, *γ* = 118.572(7)°, *V* = 1102.59 (14) Å^3^, *Z* = 4, *T* = 100.01 (10) K, space group *P*-1, *μ*(Cu K*α*) = 3.099 mm^−1^, *D_calc_* = 1.534 g/cm^3^, 3879 reflections measured (7.852° ≤ 2Θ ≤ 133.17°), 3879 unique (*R*_sigma_ = 0.0731). The final *R*_1_ values were 0.0733 (I > 2*σ*(I)). The final *wR*(F^2^) values were 0.2186 (I > 2*σ*(I)). The final *R*_1_ values were 0.1004 (all data). The final *wR*(F^2^) values were 0.2360 (all data). The goodness of fit on F^2^ was 1.020. The flack parameter was 0.18 (9) (CCDC 2221376).

### 4.7. Antibacterial Activity Assay

The antimicrobial activities against six fungi (Botrytis cinerea, Verticillium dahlia kieb., Fusarium graminearum schw., Fusarium oxysporum f.sp. niveum, Rhizoctonia solani, and Septoria nodorum Berk.) and four bacteria (Escherichia coli ATCC 25922, Staphylococcus aureus ATCC 25923, Streptococcus suis SC19, Erysipelothrix rhusiopathiae WH13013) were evaluated in 96-well plates with a twofold serial dilution method described previously [32]. Cycloheximide and penicillin were used as positive controls against fungi and bacteria, respectively.

### 4.8. Cytotoxicity Bioassay

Cell viability was analyzed by 3-(4,5)-dimethylthiahiazo (-z-y1)-3,5-di-phenytetrazoliumromide (MTT) assay as previously described [33]. In brief, cells were seeded in a 96-well plate at a density of 5 × 10^3^ per well overnight and treated with compounds for demand time. OD_570_ values were detected using a Hybrid Multi-Mode Reader (Synergy H1, BioTek, Santa Clara, CA, USA). The experiment was independently repeated three times.

### 4.9. Plate Clone Formation Assay

PC-3 cells were seeded in the six-well plate at a density of 1000 cells per well overnight, then cells were treated with DMSO (0.1 %, *v*/*v*), docetaxel (0.1 μM), compound **4** (0.16 μM, 0.32 μM, and 0.64 μM), respectively, for demand time. The cell clone colony formation was observed after two weeks of treatment. Cells were fixed with 4% formaldehyde for 30 min and washed with PBS buffer, then stained with crystal violet stain solution for 30 min. The dye solution was removed and washed the cells with PBS buffer again. Cell colonies were recorded and analyzed by the colony count analysis system (GelCount, Oxford Optronix, Oxford, UK). The experiment was repeated three times independently.

### 4.10. Apoptosis and Cell Cycle Assay

PC-3 cells were seeded in the six-well plate at a density of 2.0 × 10^5^ cells/well and incubated overnight and treated with DMSO (0.1 %, *v*/*v*), docetaxel (1.0 μM), compound **4** (2.5 μM, 5.0 μM, 10 μM), respectively, for 48 h. Then, cells were collected and stained with annexin V-FITC and PI solution, following the manufacturer’s manual (BMS500FI-300, Thermo Fisher Scientific, Waltham, MA, USA). Apoptotic rates and cell cycle distribution of PC-3 cells were examined and analyzed by flow cytometer (NovoCyte, Agilent, Santa Clara, CA, USA). Each experiment was repeated three times independently.

## 5. Conclusions

In conclusion, nine polyketides, including two new chlorinated metabolites 8-chlorine-5-hydroxy-2,3-dimethyl-7-methoxychromone (**1**) and 3,4-dichloro-1*H*-pyrrole-2,5-dione (**3**), were isolated from the mangrove-sediment-derived fungus *Mollisia* sp. SCSIO41409. The X-ray single-crystal diffraction analysis and absolute configuration of (4*S*, 5*S*, 13*R*, 14*R*, 17*R*, 18*R*, 21*R*)-stemphone C (**4**) were described for the first time. Compounds **3** and **4** showed different intensities of antimicrobial activities and antiproliferative activities. Further experiments revealed that **4** could significantly reduce PC-3 cells colony formation, induce apoptosis, and block the cell cycle at the S phase in a dose-dependent manner. Stemphone C (**4**) could be considered as a potential antiproliferative agent and a promising anti-prostate cancer lead compound.

## Figures and Tables

**Figure 1 marinedrugs-21-00032-f001:**
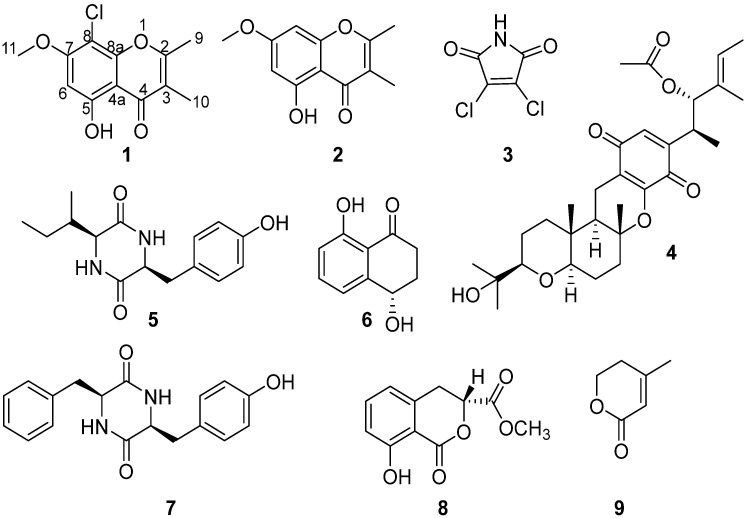
Structures of compounds **1**–**9**.

**Figure 4 marinedrugs-21-00032-f004:**
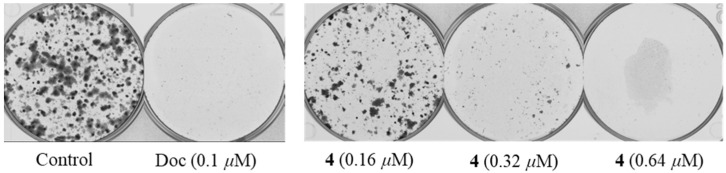
Compound **4** reduced PC-3 cells colony formation in a dose-dependent manner.

**Figure 5 marinedrugs-21-00032-f005:**
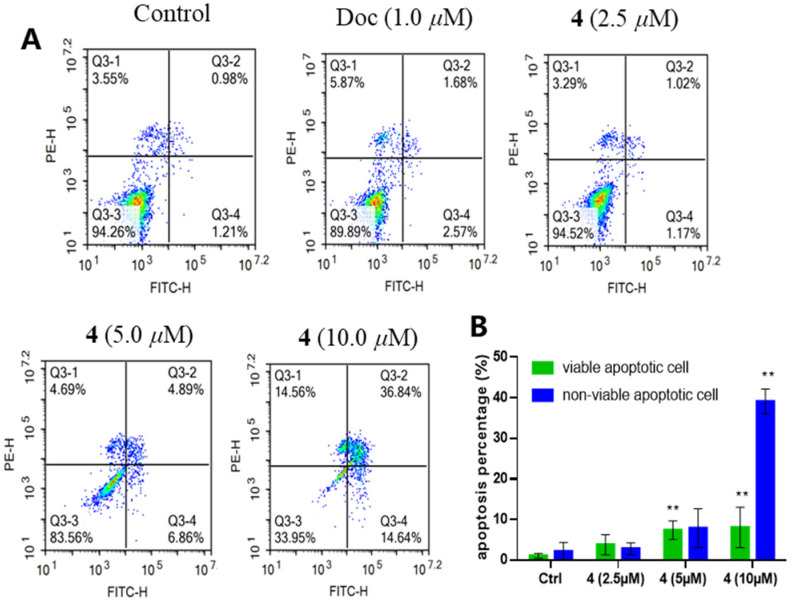
Compound **4** triggered PC-3 cells apoptosis in a dose-dependent manner (**A**,**B**). All results were presented as mean ± standard deviation (SD). Statistical significance was determined with One-Way ANOVA. ** *p* < 0.01 was considered statistically significant.

**Figure 6 marinedrugs-21-00032-f006:**
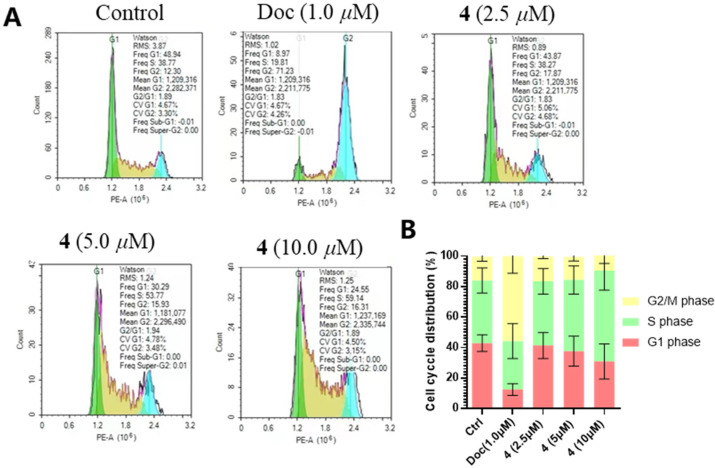
Compound **4** induced PC-3 cell cycle arresting at S phase (**A**,**B**). All results were presented as mean ± standard deviation (SD).

**Table 2 marinedrugs-21-00032-t002:** Antifungal, antibacterial, and cytotoxic activities of **3** and **4**.

Activities	Strains or Cells	3	4	Positive
Antifungal (MIC, μg/mL)	*B. cinerea*	25	>100	12.50 ^a^
	*V. dahlia*	25	>100	12.50 ^a^
	*F. graminearum*	50	>100	12.50 ^a^
	*F. oxysporum*	50	>100	>100 ^a^
	*R. solani*	50	>100	100 ^a^
	*S. nodorum*	25	50	12.50 ^a^
Antibacterial (MIC, μg/mL)	*E. rhusiopathiae*	>100	1.56	6.25^b^
	*S. aureus*	100	>100	6.25 ^b^
	*S. suis*	100	6.25	1.56 ^b^
	*E. coli*	>100	>100	50.00 ^c^
Cytotoxic (IC_50_, μM)	22Rv1	8.35	5.81	0.03 ^d^
	PC-3	9.60	2.77	0.12 ^d^
	HepG2	/	7.11	178.60 ^d^
	A549	/	11.68	29.95 ^d^
	Hela	/	11.47	/
	WPMY-1	/	5.53	0.51 ^d^
	MC3T3-E1	/	3.63	/

^a^ Cycloheximide; ^b^ Penicillin; ^c^ Streptomycin; ^d^ Docetaxel

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
