# Peer review of "New Chlorinated Metabolites and Antiproliferative Polyketone from the Mangrove Sediments-Derived Fungus Mollisia sp. SCSIO41409"

_marinedrugs, 2022, doi:10.3390/md21010032_

Round 1

Reviewer 1 Report

The manuscript entitled (New Chlorinated Metabolites and Antiproliferative Polyketone from the Mangrove Sediments-Derived Fungus Mollisia sp. SCSIO41409) by Cai et al. reported the isolation, purification, and structure elucidation of two chlorinated metabolites and eight known compounds were isolated from the mangrove sediments-derived fungus Mollisia sp. SCSIO41409. Their structures were elucidated by physicochemical properties and extensive spectroscopic analysis. The absolute configuration of stemphone C was established for the first time by the X-ray crystallographic analysis. All the obtained compounds were evaluated for the activities of pathogenic fungi and bacteria commonly found in crop plants. Compounds 3 and 4 were tested for their two human prostate cancer cell lines, PC-3 (androgen receptor negative) and 22Rv1 (androgen receptor positive).

The manuscript is good, and it could be accepted after covering the following issues

1- Different English errors should be checked and correct.

2- In introduction authors should add previously metabolites isolated from the fungus, along with their bioactivity.

Different paper on the fungus have been previously reported as:

A- Mollisianitrile, a New Antibiotic from Mollisia sp. A59Ð96.

B- New metabolites from the endophytic fungus Mollisia sp.

3- Meting point for new compounds should be measured and added in experimental section.

4- Authors evaluated all compounds as antimicrobial activites. But for cytotoxicity chose compounds 3 and 4 only, please clarify and give your reasons (Whey).

5- In supplementary material please capitalize the first letter for the name of each compound as; stemphone C should write as Stemphone C

Reviewer 2 Report

Review manuscript: marinedrugs-2108902

Title: New Chlorinated Metabolites and Antiproliferative Polyketone from the Mangrove Sediments-Derived Fungus Mollisia sp. SCSIO41409                      

1    1)    Comments: Page 2, Results:

Line 69: Compound 1: HMBC correlation from H3-9 to C-4 are long range correlation. There is no any long range correlation from H3-9 to C8a to confirm the localization of methyl groups CH3-9 at C-2? 

2   2)   Comments: Page 4, Results:

Line 107: Compound 1: On Table 1, you forget to report the long range correlation from H3-9 to C-4 as you mentioned in the manuscript.

3)      Comments: Page 6, Supporting Information:

Figure S4: is HMBC spectrum of 1 in DMSO-d6.

Figure S5: is  HSQC spectrum of 1 in DMSO-d6

Reviewer 3 Report

The manuscript “New Chlorinated Metabolites and Antiproliferative Polyketone from the Mangrove Sediments-Derived Fungus Mollisia sp. SCSIO41409reported the isolation and structural elucidation of two new chlorinated metabolites, 8-chlorine-5-hydroxy-2,3-dimethyl-7-methoxychromone (1) and 3,4-dichloro-1H-pyrrole-2,5-dione (3), together with eight known compounds (2 and 49) from the mangrove sediments-derived fungus Mollisia sp. SCSIO41409. The structure of 3 and absolute configuration of 4 were determined by X-ray spectroscopic analysis. Compounds 3 showed good antifungal and antiproliferative activities against 22Rv-1 and PC3. Interestingly, Stemphone C (4) show most significant activity on PC-3 cells, induced apoptosis, and blocked the cell cycle at S phase in a dose-dependent manner. The manuscript was prepared carefully. The results of study are sound.

I recommend the manuscript will be accepted after minor revisions.

1.      Provide the information on fermentation and extraction of fungus material.

Author Response

Response: Fermentation and extraction have been added.

“4.3. Fermentation and Extraction

The fungal strain was cultured in 200 mL seed medium (1.5% malt extract, 1.5% sea salt) in 500 mL Erlenmeyer flasks at 28 ℃ for 3 days on a rotary shaker (180 rpm). A largescale fermentation was incubated statically at 26 ℃ for 60 days using a rice medium (150 g rice, 1.5% sea salt, 150 mL H2O) in the 1 L flask (× 53). The whole fermented culture was extracted with EtOAc three times to afford a brown extract (197.2 g).”

Round 2

Reviewer 1 Report

No Comments